Machine learning based framework for fine-grained word segmentation and enhanced text normalization for low resourced language

Nazir Shahzad 1
Asif Muhammad asif@ntu.edu.pk 1
Rehman Mariam 2
Ahmad Shahbaz 1
1 Department of Computer Science, National Textile University , Faisalabad , Pakistan
2 Department of Information Technology, Government College University, Faisalabad , Faisalabad , Pakistan
Kong Xiangjie
Electronic publication date: 2024 Jan 31
Publication date: 2024
Volume: 10
Electronic Location ID: e1704
Received 2023 Aug 1; Accepted 2023 Oct 25
Copyright: ©2024 Nazir et al.
Copyright year: 2024
Copyright holder: Nazir et al.
License: This is an open access article distributed under the terms of the Creative Commons Attribution License, which permits unrestricted use, distribution, reproduction and adaptation in any medium and for any purpose provided that it is properly attributed. For attribution, the original author(s), title, publication source (PeerJ Computer Science) and either DOI or URL of the article must be cited.
License URL: https://creativecommons.org/licenses/by/4.0/

Keywords: Word segmentation, Text normalization, Machine learning, Low resourced languages

Funding: The authors received no funding for this work.

==============================
In text applications, pre-processing is deemed as a significant parameter to enhance the outcomes of natural language processing (NLP) chores. Text normalization and tokenization are two pivotal procedures of text pre-processing that cannot be overstated. Text normalization refers to transforming raw text into scriptural standardized text, while word tokenization splits the text into tokens or words. Well defined normalization and tokenization approaches exist for most spoken languages in world. However, the world’s 10th most widely spoken language has been overlooked by the research community. This research presents improved text normalization and tokenization techniques for the Urdu language. For Urdu text normalization, multiple regular expressions and rules are proposed, including removing diuretics, normalizing single characters, separating digits, etc. While for word tokenization, core features are defined and extracted against each character of text. Machine learning model is considered with specified handcrafted rules to predict the space and to tokenize the text. This experiment is performed, while creating the largest human-annotated dataset composed in Urdu script covering five different domains. The results have been evaluated using precision, recall, F-measure, and accuracy. Further, the results are compared with state-of-the-art. The normalization approach produced 20% and tokenization approach achieved 6% improvement.

Introduction

The natural language processing models (Zhou et al., 2020) require the input in a specific format to perform tasks such as text summarization (Allahyari et al., 2017), text simplification (Nisioi et al., 2017), word embedding (Nazir et al., 2022), topic modeling (Vayansky & Kumar, 2020) etc. Pre-processing is considered to be the key phase in NLP tasks (Srividhya & Anitha, 2010) and embodies two essential modules (1) normalization (Bollmann, 2019) and (2) tokenization (Hassler & Fliedl, 2006). Text normalization is a process of converting the text into scriptural standardized form of a language. While, tokenization splits the text into words or tokens. Languages such as English uses white spaces to separate the words. However, in most Asian languages, including Chinese, Thai, Urdu, and Lao, white space does not define the boundary of words (Akram & Hussain, 2010), which makes the text manipulation difficult to perform.

Urdu is an Indo-Aryan language, and previously little attention has been paid towards carrying out the natural language processing tasks for Urdu text (Daud, Khan & Che, 2017). The characters of Urdu are used to write with different variations. Similar to Arabic script (Naz, Umar & Razzak, 2016), diacritics are widely utilized in text writing. White spaces are not adequately used to separate the words. The native readers can effortlessly understand the text without spaces. However, the text processing models would not be able to recognize the boundaries of typescript. However, different normalization and tokenization approaches exist (Sharf & Rahman, 2017; Khan et al., 2022; Khan et al., 2018) to cope with such issues.

Researchers have presented different models and techniques for Urdu normalization and tokenization. The UrduHack normalization (Abbas, Mughal & Haider, 2022) serves as the state-of-the-art to normalize the Urdu text. UrduHack introduced different regular expressions for text normalization. However, the produced normalized text still lacks in tackling different aspects e.g multiple white spaces, and Urdu special characters. Likewise, for Urdu text tokenization state-of-the-art approach was introduced by Shafi et al. (2022) for text tokenization. However, the produced results are insignificant in conducting NLP experiments. This research prevails over the issues of current approaches by instigating the indispensable NLP techniques of normalization and tokenization for Urdu text. The objective of this reserach is to propose fine-grain text normalization and tokenization models for Urdu. A human-annotated large-scale dataset comprising of text from five different domains was created. For normalization, multiple regular expressions and rules were handcrafted. These rules and regular expressions addressed different aspects of text such as diacritics, normalizing characters, separating punctuation, etc. While to perform the cardinal task of tokenization supervised machine learning model was utilized with specified textual rules. The proposed normalization was evaluated using produced correct and incorrect changes in text by the approach. While, evaluation of tokenization was performed using precision, recall, F-measure, and accuracy. The produced results were compared with state-of-the-art. The normalization results were improved by 20%, while the tokenization gained 6% improvement. The language models immensely relies upon the preprocessing procedures. Ill-defined normalization and tokenization can lead to trivial results. While the explicit approaches can significantly enhance the outcomes of natural language text processing models. Therefore, the proposed text normalization and word tokenization approaches can be utilized in natural language processing tasks such as text summarization, text generation, word prediction etc. for producing high accuracy results.

Literature Review

Pre-processing serves as the key procedure to execute the text processing functions. It can be categorized into text normalization and text tokenization. For normalization and tokenization multiple approaches exist that possess their strengths and weaknesses (Shafi et al., 2022; Bollmann, 2019). These approaches are briefly described in subsequent sections.

Text normalization

The text collected from different sources can have different formats. Text normalization is deemed to transform the text into a standardized form. The pioneering approach for text normalization was proposed by Baron & Rayson (2008) dealing with spelling variations. The authors presented the Variant Detector tool for identifying different spellings for the same word and served as a pre-processor. The Urdu alphabet and numeric characters are presented in Table 1.

Table 1 Urdu alphabets and numbers.

	

An approach for the normalization of the text of two Asian languages, such as Urdu and Hindi, was proposed by Mehmood et al. (2020). The authors focused on Roman Urdu and Roman Hindi language as their writing script is very much similar. The author performed lexical normalization on text, and this process involved mapping spelling variations to a specific word that can be considered a standard form. This improved the performance of text mining and natural language processing tasks while reducing the sparsity of data instances.

Further, to convert the words into a standardized form, Sproat & Jaitly (2016) proposed different approaches based on RNN (Khaldi et al., 2023) for normalizing the text. The dataset collected for this research consisted of 290 million words in the Russian language and 1.1 billion words in the English language, which were extracted from Wikipedia. From the test dataset, manual analysis was performed on 1000 examples, and approximately the error rate for the English language was observed as 0.1%. The error rate for the Russian language was 2.1%, which was more than the error rate for English. For the English language, 1,000 unique tokens were permitted, while for the Russian language, 2,000 words were allowed. The model LSTM used the RNN model with single input and output layers. The language model and channel outputs were combined at the time of decoding. The model achieved 99% accuracy for the English language and 98% accuracy for the Russian language.

Clark & Araki (2011) proposed an approach for text normalization that normalizes the text on social media. With a rapid increase in the usage of social media platforms, vast amounts of data are being generated daily. This form of data is usually in amorphous form, and natural languages processing processes such as information retrieval, machine translation, and opinion mining are not feasible to apply directly. The error word represents the simple English word, and the regular word is the corresponding word in the English dictionary. The phase matching rules were applied to the text, which is essential for text analysis. For Urdu text normalization, the Urdu Hack (Abbas, Mughal & Haider, 2022) library is considered as the state-of-the-art for performing such a task. It has been utilized in significant research works. However, this work lacks in multiple aspects as it does not handle multiple white spaces, Urdu special characters, i.e., Therefore, there is need to enhance the existing work.

Text tokenization

Tokenize the text is a pivotal phase, as the reader needs to be split into separate words so the machine can understand the language (Grefenstette, 1999). Similarly, for the tokenization process, there are different approaches as well. To solve the tokenization problem, Hassler & Fliedl (2006) proposed a rule base model that extended the tokenization from splitting to domain knowledge preservation. The key features were identifying linguistical markers and their disambiguation, identifying and converting abbreviations into their full form, and tackling different formats. To improve text mining quality, tokenization based on linguistics was proposed as a compulsory text processing job.

A tokenizer was introduced by Zhang, Li & Li (2021). The authors proposed A Multi-grained BERT (AMBERT) based on coarse-grained and fine-grained tokenization of text. The parameters were shared between the encoders, and the output of both encoders was combined to form the final result for tokens and phrases. To tokenize the Urdu text, Durrani & Hussain (2010) proposed a model based on linguistic and orthographic features. The author stated that in the Urdu language, space is not compulsory for separating words. Instead, the readers can identify the boundaries of words even without the presence of proper space among words. Space is considered for assigning appropriate shapes to words. Such as (cells tissues) are two words and contain a space between them. However, by removing space between the words , these words would still be readable and visually correct. While (proper quantity) is valid with the presence of white space, however, if we omit the white space, the words would be merged and would become visually incorrect. A character based on shape can have four different variants such as (1) at the start of a word, (2) in the middle of a word, (3) at the end of a word, and (4) isolated. The characters of the Urdu language can be categorized into two classes, joiners, and non-joiners. The joiners and non-joiners for the Urdu language are presented in Fig. 1.

Figure 1 Joiners and non-joiners urdu alphabets.

The issues faced during the tokenization of Urdu text are (1) space omission (Lehal, 2010), (2) space insertion (Lehal, 2009), (3) affixation (Khan, 2013), and (4) reduplication (Afraz, 2012). Table 2 presents the space omission problem.

Table 2 Space omission with Urdu text.

	

A sentence is a stream of words that convey some thought and contains a predicate and subject. A subject can be a single word or a compound word. It provides information about whom or what in a sentence. However, in the Urdu language, challenges exist because of its ambiguous sentence boundary. A hybrid approach was introduced by Shafi et al. (2022). The authors proposed two different methods to the tokenization of Urdu text. The first approach was based on rules. The author used five different rules for the identification of Urdu word boundaries. Another method based on a machine learning model was also proposed, considering the support vector machine algorithm. Various features were considered for incorporation in the support vector machine algorithm. The accuracy achieved by the model was 0.92. While the recall value was 0.87, the precision was 0.91, and the achieved F-measure was 0.89. However, the results produced by state-of-the-art are not significant for performing NLP tasks.

Methodology

This research is conducted to propose Urdu text normalization and tokenization models. The key phases are dataset creation, defining normalization rules and regular expressions, defining tokenization features, classifying features, predicting character sequences, evaluating results, and results comparison. The overall methodology is presented in Fig. 2.

Figure 2 Overall proposed methodology.

Dataset creation

This experiment is performed with contents from five domains written in the Urdu language were considered, such as (1) biology, (2) physics, (3) chemistry, (4) Urdu literature, and (5) social studies. However, the contents were in scanned form instead of text. The text extraction was performed by utilizing Google Lens, a powerful application for extracting text from images (Shapovalov et al., 2019). While the extracted text was partially broken, incomplete, and altered. To annotate the dataset, three domain experts were hired. The annotators reconstructed the incomplete and broken words and introduced proper spacing between the words and other symbols. The text of all five domains was further combined to form the final dataset. The dataset sample is presented in Fig. 3. It can be observed from the figure that words are properly spaced. The other symbols are also separated from the text.

Figure 3 Urdu dataset.

The statistics of the dataset are presented in Table 3. The combined dataset is consisted of 158,351 words. This is the largest dataset that is created with proper spacing for performing NLP tasks till now.

Urdu text normalization

Normalization is an essential phase to perform NLP tasks for the Urdu language. The basic issues while performing language-based experiments are coped with normalization. It converts the input text into its basic and original form and enhances the efficiency of machine learning jobs. For normalization, we introduce the following rules.

Removing diacritics

In NLP models, the diacritics can reduce the output to some extent. The word with a diacritic and the same without a diacritic would be considered different. Therefore, to remove the diacritics from Urdu text, we brought the characters in the range 0x0600–0x06ff (Khan et al., 2021). Equations (1)–(2), presents the rule for removing the diacritics. (1) text=sentencefromcorpus

(2) new_text=′.jointfortintextifftnotinunichrxforxinrange0x0600,0x06ffiffunicodedata.categoryunichrx==′Mn′

Table 3 Statistics of all datasets.

Sr#	Domains	Words	
1	Social studies	25,763	
2	Urdu literature	27,684	
3	Biology	37,176	
4	Physics	35,763	
5	Chemistry	31,965	
6	Combined dataset	158,351	

Separating punctuation and sentence ending characters

For this purpose, we utilized the string library and downloaded the set of punctuation characters (Martín-del Campo-Rodríguez et al., 2019). To standardize Urdu text, we introduced a white space after each punctuation mark and removed the white space before punctuation. The rule for separating punctuation and sentence ending characters is given in Eqs. (3) and (4). (3) s=setofUrdupunctuation/sentenceendingcharacters

(4) ∀x∈s,iffxintext;new_text=text.replacex,x+′′

Similarly, the a set of ending sentence ending characters was created and if that specific character occurred in text space before it was removed and single space after such character was introduced.

Separating digits

The digits should be separated from the text; otherwise, the natural language processing models would consider the digits and joined text as a single word. Therefore, we placed a white space before and after the digits. We have also incorporated the ‘.’ symbol between the digits. If it appears between the digits, no white space will be placed. (5) new_text=re.subr′0−9+.0−9+?′,r′1”,text

In Eq. (5), regular expression has been provided for separating digits from other text.

Separating special, mathematical and greek symbols

In Urdu text, mathematical and Greek symbols are also widespread. To separate these symbols from the text, we introduced white spaces before and after such characters as presented in Eqs. (6) and (7). A rule was also developed to separate special that are explicitly used in the poetic text. (6) s=setofspecial/mathematical/greekcharacters

(7) ∀x∈s,iffxintext;new_text=text.replacex,′′+x+′′

Separating English words

In Urdu texts, English words are commonly observed. These words can convey meaning whether they are separate from Urdu text. However, in text processing, this can affect the final results. Therefore, white spaces were placed before and after English words. (8) new_text=re.sub′A−Za−z+′,lambdaele:′′+ele0+′′,text

The Eq. (8) represents the regular expression for separating the English words from other text.

Standardizing characters

Urdu writing script is complex to analyze. A single character can have different shapes of writing, such as are various shapes of the same character. This can negatively affect the results. We converted the variations into standardized forms by developing the appropriate rule presented in Eqs. (9)–(11). (9) s_set=standardizedUrducharacters

(10) v_set=setsofUrducharactervariations

(11) ∀s∈s_set,v∈v_setiffviintext:replacevwiths

Removing extra spaces

Urdu text can contain extra white spaces during composition; similarly, while implementing prior rules and regular expressions, extra spaces are possible in the text. To tackle this issue, we developed a regular expression to remove extra white spaces. Using regular expression as presented in Eq. (12), more than white spaces would be removed from text. (12) new_text=′re.sub∖s∖s+′,′′,text

The proposed rules and regular expressions with possible outcomes are presented in Table 4. The regular expressions and rules were created to normalize the text and convert it into standard Urdu text. The rules were manually extracted (Shafi et al., 2022) with the help of domain experts. These expressions can significantly increase the effective manipulation of text while reducing the computation complexity if we do not remove the diacritics. The model will consider both words differently.

Table 4 Proposed rules and regular expressions for Urdu text normalization.

	

Tokenization

Text tokenization is the task of identifying the word streams in a written corpus (Webster & Kit, 1992). It splits the text into tokens or words. It is a compulsory job for any language processing experiment, such as speech tagging, machine translation, information extraction or retrieval, etc. Tokenization can be reflected in the following phases.

Defining key features

After performing manual annotation, the next phase would be based on feature extraction. These tasks need input written text with clear boundaries. Sentence splitting is separating sentences based on punctuation in the text. Tokenization and sentence splitting are complex tasks when it comes to the Urdu language, as there are irregular white spaces between words. The features useful for tokenization would be extracted from the text corpus and forwarded to the machine learning algorithm. The features can be the following:

1. Current character of text

2. Previous seven characters from the current character

3. Next seven characters from the current character

4. Identification, if the current character is non-joiner

5. Identification, if the current character is a joiner

6. Identification, if the current character is a digit

7. Identification, if the current character is a Greek letter

8. Identification, if the current character is a mathematical character

9. Identification, if the current character is a symbol

10. Identification, if the current character is from the English language

11. Unicode class of the current character

Feature classification

The native speaker can identify the word boundaries. However, for machines, it is difficult. The characters in Urdu text can adopt different shapes when joined based on context. A character can be categorized as (1) starting, (2) ending, (3) middle, and (4) separated. Such characters are termed a joiner, while on the other hand, if a character can make a posse, only two forms, ending and separation, are referred to as non-joiners. When two morphemes join while forming a word, the writer would like inert a space for better visualization if the initial morpheme ends with a joiner. On the other hand, if the first morpheme ends with a non-joiner, the writer would not inset white space because there would be no effect on shape or word. The sub-words are separated with zero-width non-joiner.

For accurately performing the tokenization of Urdu text, we utilized the conditional random field (CRF) algorithm (Lafferty, McCallum & Pereira, 2001), which used different linguistic features for the identification of white space as the boundary of a word and zero-width non-joiner (ZWNJ) as a boundary of sub-word. CRF is a type of probabilistic graph model that considers the neighboring context for performing different tasks such as classification. In CRF the dependencies are implemented between the predictions. CRF was considered to specify the word boundaries. However, We have also developed rules for separating punctuation, mathematical symbols, special characters, and English words and removing more than single spaces. The linear chain CRF can be expressed with Eq. (13). (13) Py|x= ∑hPy|h,xPh|x

where the observations are x = x1, x2.... xn and labels are y = y1, y2, …. yn. The set of latent variables is presented by h. The complete code of normalization and tokenization with dataset have been uploaded on GitHub (https://github.com/Shahzad-Nazir/Normalization-and-Tokenization). The research community popularly uses this model for conducting NLP experiments (Geng, 2021). For training, the model, 60% dataset, and for testing, 40% dataset was utilized (Nazir et al., 2020).

Predicting sequence of character

To accurately perform the tokenization, the CRF linear model was trained on a given set of features against each character of text. The stream of characters was converted into 0s and 1s. If the current character is the last character of a word, it was assigned a label 1, and for other characters, 0 was assigned. The label 1 denotes that the next would be white space, while 0 indicates that the next would be a non-space character. After training, the model could predict the appropriate sequence of characters of words.

Evaluation metrics

The normalization is evaluated by computing the correct and incorrect changes introduced by the model. The correct changes would be the alterations necessary to normalize the text. Conversely, the incorrect changes would be erroneous alterations in normalization of text. While, for evaluating word tokenization, the performance measures such as precision, recall, F-measure and accuracy would be considered (Sokolova, Japkowicz & Szpakowicz, 2006). These measures can be calculated as shown in Eqs. (14)–(17). (14) Precision=TPTP+FP

(15) Recall=TPTP+FN

(16) F.measure=2∗Precision∗RecallPrecision+Recall

(17) Accuracy=TP+TNtotalinstances.

Precision is considered as a key measuring to examine the quality of classification. The higher precision value indicates that the algorithm has returned less irrelevant and more relevant results. While the completeness is represented by the value of recall. The value of precision is computed with ratio of the retrieved relevant articles and all number of articles. In contrast, the value of recall is calculated through dividing the retrieved relevant articles by the total articles. Further, such measures are computed through (1) true positive (TP), (2) true negative (TN) (3) false positive (FP) and (4) false negative (FN).

Here, TP would be the character in a word that was a non-ending and model predict it correctly. FP would be the character that was ending character but predicted as non-ending character. Likewise, TN refers to the ending character which was predicted as ending character. While FN would be the ending character but predicted as non-ending.

Results and Discussions

Pre-processing on textual data is considered as a significant task while performing NLP research works (García et al., 2016). Normalization and tokenization are two imperative actions of pre-processing phase. The normalization converts the text into the standardized script, and for Urdu normalization we proposed ten regular expressions such as (1) separating punctuations, (2) removing diacritics, (3) separating digits, (4) separating English words from Urdu text, (5) converting varying character shapes into standardized, (6) separating Greek characters, (7) separating mathematical symbols, (8) separating Urdu digits, (9) separating specific poetic letters, (10) removing uneven spaces. The separation of punctuation and other symbols is a basic need. If any punctuation numeric text is without a space with Urdu words, the punctuation, and the word would be considered a single word. In NLP ‘ ’ are two different words as a single space is present between them. If we remove the space between the words ‘ ” both words would be considered as a single entity. The proposed expressions were applied to five Urdu domain contents separately. The normalization results produced by expressions are presented in following Table 5.

Table 5 Normalization results achieved by proposed rules and regular expressions.

Dataset	Correct changes	Incorrect changes	Remained changes	
Biology	1,922	12	0	
Physics	2,213	3	0	
Chemistry	1,798	22	0	
Urdu literature	2,079	27	0	
Social studies	1,548	40	0	
Combined dataset	9,560	104	0	

The proposed rules and regular expressions made 1,922 correct changes, 12 incorrect changes, and converted the Biology text into the normalized form. Similarly, correct changes were made for physics and chemistry (2,213 and 1,798, respectively). For Urdu literature and social studies, 2,079 and 1,548, respectively, changes occurred. While for the combined dataset, the changes in text were observed to be 9,560. The proposed rules and regular expressions covered all the normalization aspects; therefore, the remaining changes were 0 in all datasets. However, incorrect changes were observed. These incorrect changes occurred due to the existence of commas between numeric values. As the spaces were inserted before and after numeric data, the commas between the digits were not properly normalized. For example, in normalized form, the consecutive digits appeared as “24, 28, 29” instead of “24, 28, 29”. Similarly, the punctuation after closing small parenthesis were not normalized. The proposed regular expressions converted the text into the normalized form.

Likewise, word tokenization is the key process in text manipulation. This procedure splits the whole text into words termed as tokens. These tokens are further fed into machine learning models such as BERT, Transformer, Encoder-Decoder, etc. for performing NLP tasks. The dataset of five different domains was used for Urdu tokenization, containing properly spaced Urdu text. Against each character 11 features were identified. From the current character, different window sizes were investigated and high results producing window size of seven next and previous characters was selected. Further, the stream of text was labeled with 0’s and 1’s. If the character is initial or middle, it was labeled with 0, while the ending character was labeled as 1. Here, 1 depicts that after this character a white space exists. The whole dataset was split into 60% and 40% for training and testing. The conditional random field (CRF) linear model was trained on 60% of the dataset. The model learned to introduce the white space where it is required. The trained model was tested on 40% of the data. Specifically, in testing the model itself predicted and introduced the white spaces. The results were evaluated using precision, recall, F-measure, and accuracy (Goutte & Gaussier, 2005) and other measures that are presented in Table 6.

Table 6 Evaluation measures for each dataset.

Domain/Dataset	Precision	Recall	F-measure	Accuracy	
Physics	0.9976	0.9965	0.9971	0.9958	
Biology	0.9947	0.9946	0.9947	0.9921	
Chemistry	0.997	0.996	0.9965	0.9948	
Urdu literature	0.9876	0.9886	0.9881	0.9835	
Social studies	0.992	0.9906	0.9913	0.9872	
Combined data	0.9942	0.9946	0.9944	0.9919	

For biology, physics, and chemistry, the model achieved 0.99 accuracy, while for Urdu literature and social studies, the model achieved 0.98 accuracy. Similarly, for the combined dataset, 0.98 accuracy was achieved. These results are based on actual positive, true negative, false positive, and false negative values. These values for each dataset are presented in Table 7.

Table 7 Evaluation of results for urdu text tokenization on datasets.

Sr#	Books/Dataset	True positive	False positive	True negative	False negative	
1	Physics	88,941	212	34,772	313	
2	Chemistry	86,260	262	31,072	348	
3	Biology	101,731	541	35,849	549	
4	Social studies	68,591	556	24,533	653	
5	Urdu literature	61,417	771	26,510	707	
6	Combined dataset	407,052	2,370	152,910	2,219	

The table presents the predicted labels for each character in the datasets. It can be observed that most of the predicted labels are true positives or true negatives. The values of false positives and false negatives are very avoidable. The combined dataset contains 407,053 true positive labels, 2,370 false negative labels, 152,910 true negative, and 2,219 false negative labels. The model can significantly tokenize the Urdu text and precisely predict the text’s white space. It can tokenize text that is not readable by native individuals.

The tokenization model is presented in Fig. 4, where it can be observed that the model can appropriately introduce the white space in the text where it is needed. The first line in the figure is the text without any space and was fed to the model, and the proposed model perfectly tokenized the whole unreadable text.

Figure 4 Fine-grained tokenization of Urdu text.

Comparison

To conduct the normalization process, we introduced 10 rules and regular expressions to normalized the Urdu text. While the state-of-the-art approach is rule-based that proposed six regular expressions for the normalization of Urdu text and is termed UrduHack normalization (Abbas, Mughal & Haider, 2022). The proposed six rules by the state-of-the-art are (1) removing diacritics, (2) normalizing single characters, (3) normalizing compound characters, (4) placing space after punctuations, (5) placing space before and after English words, (6) placing space before and after digits. The datasets of five domains were fed to a state-of-the-art approach, and UrduHack normalization for biology was able to make 1,598 correct changes, 150 incorrect changes, and 162 remaining changes. The maximum number of correct changes were produced for a dataset of physics, covering 2,115 correct changes, 52 incorrect changes, and 43 remaining. However, for the combined dataset total of 7,928 correct changes were made, incorrect changes were 701, and the remaining changes were observed to be 827. Table 8 illustrates the results of UrduHack. The comparison of UrduHack and the proposed model is presented in Figs. 5 and 6.

Table 8 Results achieved by urduhack on all datasets.

Dataset	Correct changes	Incorrect changes	Remained changes	
Biology	1,598	150	162	
Physics	2,115	52	43	
Chemistry	1,596	101	79	
Urdu literature	1,709	117	226	
Social studies	910	281	317	
Combined dataset	7,928	701	827	

Figure 5 Results comparison for correct changes.

Figure 6 Results comparison for incorrect changes.

The proposed regular expressions performed better as compared to the state-of-the-art approach. The state-of-the-art approach could not accommodate essential characters, such as simple quotation marks, double quotation marks, numeric operators, etc. Similarly, special characters were also not considered. A major drawback of UrduHack is not handling irregular spaces. If a white space is present with punctuation, it will insert one more white space. The proposed approach produced 20% better results than the recent state-of-the-art approach.

Multiple supervised and unsupervised approaches exist for word tokenization. The state-of-the-art UNLT-WT approach was introduced by Shafi et al. (2022), which utilized 59,000 tokens and produced 0.92 accuracy using SVM with grammatical rule. Therefore, we also considered the dataset of state-of-the-art used for word tokenizer and was fed to our model. The model produced 0.98 accuracy, as presented in Fig. 7.

Figure 7 Results comparison for tokenization.

The results of Shafi et al. (2022) were further explored, and the confusion matrix was obtained. The true positives were 202,729, the false positives were 2,530, the false negative were 933, and the true negative characters were observed to be 2,581. The proposed model obtained higher results as compared to the state-of-the-art. The proposed approach can be utilized for Urdu text tokenization.

Conclusion

The text preprocessing is considered as the key phase in NLP tasks such as text summarization, next word prediction, text generation, etc. Text normalization and word tokenization are two essential modules, while performing text preprocessing. There exist well developed preprocessing models for most spoken languages. However, towards low-resourced languages such as Urdu, researchers have paid minor attention. This research presents two text pre-processing modules: normalization and tokenization for low-resourced Urdu. A dataset consisting of contents form five domains in Urdu script was created to perform this research. The raw text was normalized with 10 rules, and regular expressions, such as standardizing single characters, removing diacritics, separating punctuation, separating digits, etc. While, to tokenize the text, we utilized the conditional random field (CRF) linear model with specified grammatical rules. For word tokenization, the dataset was split into training and testing datasets. Training was performed on 60% dataset. While for testing, 40% dataset was considered. Against each character of text 11 features were extracted and fed into the machine learning model CRF. Further, specific rules were applied. Precision, recall, F-measure, and accuracy were considered for results evaluation. The normalization results were improved by 20%. The word tokenization model was able to achieve 0.98 accuracy score. The tokenization results were improved by 6% as compared to the state-of-the-art approach.

Additional Information and Declarations

Competing Interests

Author Contributions

Data Availability

Muhammad Asif is an Academic Editor for PeerJ Computer Science.

Shahzad Nazir conceived and designed the experiments, performed the experiments, analyzed the data, performed the computation work, prepared figures and/or tables, authored or reviewed drafts of the article, and approved the final draft.

Muhammad Asif conceived and designed the experiments, performed the experiments, analyzed the data, performed the computation work, prepared figures and/or tables, authored or reviewed drafts of the article, and approved the final draft.

Mariam Rehman conceived and designed the experiments, performed the experiments, analyzed the data, performed the computation work, prepared figures and/or tables, authored or reviewed drafts of the article, and approved the final draft.

Shahbaz Ahmad conceived and designed the experiments, performed the experiments, analyzed the data, performed the computation work, prepared figures and/or tables, authored or reviewed drafts of the article, and approved the final draft.

The following information was supplied regarding data availability:

The complete project code along with data set is available at GitHub and Zenodo:

- https://github.com/Shahzad-Nazir/Normalization-and-Tokenization

- Shahzad Nazir. (2023). Urdu Text Normalization and Tokenization Dataset (1.0) [Data set]. Zenodo. https://doi.org/10.5281/zenodo.8372388.

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
