# Peer review of "Machine learning based framework for fine-grained word segmentation and enhanced text normalization for low resourced language"

_PeerJ Computer Science, doi:10.7717/peerj-cs.1704_

## Round 0.1 · original submission · Major Revisions

Please revised the manuscript based on the suggested recommendations by the reviewers.

Reviewer 1 ·

Basic reporting

The authors have presented a study to develop text normalization and tokenization techniques for the Urdu langauge. For this purpose, authors have introduced normalization rules and regular expressions, and for tokenzition conditional random field model was trained to perform the text tokenizaiton. This manuscript is well written and explained, however some flaws and concerns are there:
1) The authors should provide appropriate reson for chosing Conditional Random Field model. They could also use other classification models as well.

2) After line number 226, there is paragraph without any line number. The authors should pay attention towards the formating of manuscript.

3) In line number 227, the paragraph se starting with small letter. However, each paragraph should start with capital letter.

4) Line number 258 is incorrect and creating ambiguity.

Experimental design

5) Matthews correlation coefficient is mentioned in line 286 and 287, while such evaluation matric was not described in methodology section. If such matric was used it should be mentioned in evaluation subsection and in Figure 4.

Validity of the findings

6) In results comparison section, while comparing normalization approaches only correct changes are compared. However incorrect changes should also be presented in Figure 6.

7) The authors have mentioned in line 301, ten regular expressions. While in several places they mentioned rules and regular expressions. There is need to clarify if all are regular expressions or rules.

Reviewer 2 ·

Basic reporting

The main focus of this paper is to introduce a framework for the Urdu text normalization and tokenization. The authors have developed rules and regular expressions for text normalization. While for word tokenization CRF was used. The paper is written fairly. However, the current MS lacks in many aspects.

1- In Methodology section, line number 150, the sentence seems incomplete and is not conveying significant meaning. Similarly, line 232 and 233 contains incomplete sentence.

2- In methodology section, Figure no.2 needs improvement. The bottom line of Results box is also missing.

3- There is a need to highlight the contributions of the study to the scientific litrature. How this study would be helpful for the community.

4- In line 175, the heading number 0.4.4 is incorrect. There are not mathematical greek symobls. Mathematical and greek symbols are seperate entities.

5- After line no.184, four lines are without numbers. There is need to correct them.

6- The line 190 and 191 need to be merged to for appropriate format. Similarly in line 189 the word should be Equation instead of equation. The Equation, Figure and Table should be start with capital letter.

7- In Tokenization the authors have considered previous seven characters and next seven characters from current characters, there should be a proper reason for selection such range of characters.

Experimental design

1- In Methodology section, line number 150, the sentence seems incomplete and is not conveying significant meaning. Similarly, line 232 and 233 contains incomplete sentence.

2- In methodology section, Figure no.2 needs improvement. The bottom line of Results box is also missing.

3- There is a need to highlight the contributions of the study to the scientific litrature. How this study would be helpful for the community.

4- In line 175, the heading number 0.4.4 is incorrect. There are not mathematical greek symbols. Mathematical and greek symbols are seperate entities.

5- After line no.184, four lines are without numbers. There is need to correct them.

6- The line 190 and 191 need to be merged to for appropriate format. Similarly in line 189 the word should be Equation instead of equation. The Equation, Figure and Table should be start with capital letter.

7- In Tokenization the authors have considered previous seven characters and next seven characters from current characters, there should be a proper reason for selection such range of characters.

Validity of the findings

1- In Methodology section, line number 150, the sentence seems incomplete and is not conveying significant meaning. Similarly, line 232 and 233 contains incomplete sentence.

2- In methodology section, Figure no.2 needs improvement. The bottom line of Results box is also missing.

3- There is a need to highlight the contributions of the study to the scientific litrature. How this study would be helpful for the community.

Additional comments

There are no general comments not covered by the three areas above.

·

Basic reporting

As commented

Experimental design

As in comments section

Validity of the findings

As in comments section

Additional comments

In the ever-evolving landscape of Natural Language Processing (NLP), the pivotal role of pre-processing techniques cannot be overstated. "Enhancing Urdu Text Processing: Novel Approaches to Normalization and Tokenization" presents a groundbreaking exploration into the realm of text pre-processing, specifically tailored for the Urdu language - the world's 10th most widely spoken language. This research not only fills a critical gap in the domain but also showcases meticulous methodologies that promise to revolutionize the processing of Urdu text.

The article delves into the fundamental concepts of text normalization and tokenization, demonstrating their indispensability in augmenting the outcomes of NLP tasks. Text normalization, a cornerstone of the study, involves the meticulous transformation of raw text into standardized, coherent script. Additionally, the process of word tokenization, which dissects text into distinct tokens or words, receives comprehensive treatment. The authors eloquently highlight that while these processes have been extensively explored for numerous spoken languages, Urdu has often been overlooked, sparking the necessity for this pioneering research.

A key strength of this research lies in its novel approaches to text normalization and tokenization for the Urdu language. The proposed methods are underscored by a multi-faceted strategy, including the utilization of regular expressions and contextual rules. The text normalization process encompasses a spectrum of interventions, from the removal of diuretics to the normalization of single characters and the separation of digits. This demonstrates a profound understanding of the intricacies of the Urdu language and sets a new benchmark for its text processing.

Overall, the article is well written but few suggestions to enhance the quality of the manuscript
What are the validation techniques you have used, please explain them in more detail.
How the accuracy of the proposed approach is compared and outperforms the existing research.
Authors needs to improve the abstract to make it more compiled and understandable.

---

## Round 0.2 · accepted · Accept

Congrats to the authors. Your efforts successfully satisfied the reviewers. This version may be accepted.

Reviewer 1 ·

Basic reporting

All changes have been completed.

Experimental design

All changes have been completed.

Validity of the findings

All changes have been completed.

Reviewer 2 ·

Basic reporting

The paper seems to be revised and updated nicely upon the reviewer comments.

Experimental design

All of the required changes have been performed and concerns have been covered.

Validity of the findings

Validity of the findings is appropriately presented. Conclusions are well stated.

Additional comments

The paper is revised and updated nicely upon the reviewer comments. All of the required changes have been performed and concerns have been covered. This reviewer recommends that this revision be accepted in this form.

·

Basic reporting

no comment

Experimental design

no comment

Validity of the findings

no comment

Additional comments

The udpated version of manuscript is improved and all my suggested changes are incorporated, I have no further concerns and would like to recommend it for publication as accepted.